# Effect of Nitrogen Doping on the Performance of Mesoporous CMK-8 Carbon Anodes for Li-Ion Batteries

**G. Calcagno** [1,*] , **M. Agostini** [2], **S. Xiong** [2], **A. Matic** [2], **A. E. C. Palmqvist** [1] **and C. Cavallo** [2,3,*]

1    Department of Chemistry and Chemical Engineering, Chalmers University of Technology,
     412 96 Gothenburg, Sweden; anders.palmqvist@chalmers.se
2    Department of Physics, Chalmers University of Technology, 412 96 Gothenburg, Sweden;
     agostini@chalmers.se (M.A.); shizhao.xiong@chalmers.se (S.X.); matic@chalmers.se (A.M.)
3    Centre for Materials Science and Nanotechnology, Department of Chemistry, Oslo University,
     0371 Oslo, Norway
*    Correspondence: giulio.calcagno@chalmers.se (G.C.); carmen.cavallo@smn.uio.no (C.C.)

**Abstract:** Designing carbonaceous materials with heightened attention to the structural properties such as porosity, and to the functionalization of the surface, is a growing topic in the lithium-ion batteries (LIBs) field. Using a mesoporous silica KIT-6 hard template, mesoporous carbons belonging to the OMCs (ordered mesoporous carbons) family, namely 3D cubic CMK-8 and N-CMK-8 were synthesized and thoroughly structurally characterized. XPS analysis confirmed the successful introduction of nitrogen, highlighting the nature of the different nitrogen atoms incorporated in the structure. The work aims at evaluating the electrochemical performance of N-doped ordered mesoporous carbons as an anode in LIBs, underlining the effect of the nitrogen functionalization. The N-CMK-8 electrode reveals higher reversible capacity, better cycling stability, and rate capability, as compared to the CMK-8 electrode. Coupling the 3D channel network with the functional N-doping increased the reversible capacity to ~1000 mAh·g$^{-1}$ for the N-CMK-8 from ~450 mAh·g$^{-1}$ for the undoped CMK-8 electrode. A full Li-ion cell was built using N-CMK-8 as an anode, commercial LiFePO$_4$, a cathode, and LP30 commercial electrolyte, showing stable performance for 100 cycles. The combination of nitrogen functionalization and ordered porosity is promising for the development of high performing functional anodes.

**Keywords:** ordered mesoporous carbon (OMC); CMK-8; nitrogen doping; functional materials; lithium-ion batteries (LIB)

## 1. Introduction

As the market of electromobility and energy storage expands, batteries with higher capacity, cyclability, and safety are deemed necessary [1]. Non-aqueous rechargeable Li-ion batteries (LIBs) are one of the most prominent energy storage devices, and their development represents a prodigious success of modern materials electrochemistry that led to the Nobel Prize of Chemistry in 2019.

Replacing lithium metal with cheaper and more abundant anodes such as sodium (Na), potassium (K), calcium (Ca), magnesium (Mg), aluminum (Al), etc., has become a valid alternative to lithium metal batteries (LBs). However, when it comes to metal-ion batteries (MIBs), other systems are still not competitive with Li-ion systems [2,3].

Currently, LIBs are prevalent when we think about the battery market for portable electronic devices. However, this technology has also been successfully implemented in hybrid electric vehicles (HEVs) or battery electric vehicles (BEVs), as well as in stationary energy storage [4]. All these systems

are running with a graphite-based anode material. Unfortunately, graphite, due to its low specific capacity, cannot meet the growing request for small electronics, electric cars, and electricity storage stations [5].

Graphitic and non-graphitic carbon materials are recognized as some of the most interesting candidates for anode materials [6], both for Li, Na-ion batteries, and other energy storage applications [7–13], due to their high electrical conductivity, high chemical and thermal stability, and large surface area, low cost and easy availability [14]. Carbon spheres [15], carbon nanotubes [16], as well as carbon nanofibers [17] and graphene [18–20], constitute a new plethora of possible replacements for graphitic carbon.

In the past 10 years, ordered mesoporous carbons (OMCs), which consist of a family of materials called CMK-n, have gained more scientific interest from the point of view of anode lithium battery materials. This is mainly due to their unique structural and morphological characteristics. Ordered mesoporous interconnections together with high surface areas, large pore volumes, and uniform pore size distribution are the most interesting properties of this class of materials [21]. Properties that provide short pathways for ion transfer, better electrolyte absorption, and higher specific capacity, paving the way for an increase in performance in the final device [22].

OMCs are commonly synthesized via a nanocasting method using mesoporous silica materials as hard templates. Highly-used templates are named MCM-48, SBA-1, SBA-15, and SBA-16 [23,24]. The chemistry literature reports CMK-3, a reverse replica of the SBA15, as the most used battery OMCs carbon material [25]. On the other hand, less attention has been paid to CMK-8, even if its interpenetrating bicontinuous network of channels is a tempting feature as pathways for Li-ions. The three-dimensional (3D) cubic mesoporous carbon, CMK-8, can be prepared through a reverse replica of 3D cubic mesoporous silica KIT-6 (Ia3d symmetry) [26].

In recent years, different functionalization [27], doping [28], and composite [29] have been used to improve the electrochemical performance of several carbon materials [30]. Nitrogen functionalized/doped materials are the most attractive class due to the nitrogen electronegativity (3.5), which is higher than that of carbon (3.0) and atomic diameter, which is smaller. Therefore, stronger interaction between the nitrogen-doped carbon material and lithium ions is reasonably predictable for an easier lithium insertion/extraction [31].

This study demonstrates that functionalizing/doping mesoporous carbon (N-CMK-8) with nitrogen can serve as highly reversible anode material for LIBs, with superior capacity and cycling stability. Besides, to study the anode side of a whole lithium-ion cell, we report a formulation of a full device with a commercial LFP as a cathode material. The N-doped mesoporous carbon having large surface area and uniform pore width facilitates the contact between electrode and electrolyte and the transport of electrolyte ions. Besides, the N-doping generates defect sites and provides many active sites for Li-ion storage. All these characteristics are propitious for high specific capacity and excellent performance of N-CMK-8 for LIBs.

## 2. Materials and Methods

### 2.1. Synthesis Process

#### 2.1.1. Silica Template KIT-6

The mesoporous cubic Ia3d silica (KIT-6) was produced in an acidic aqueous solution using a mixture of Pluronic P123 ($EO_{20}PO_{70}EO_{20}$, MW: 5800 g/mol), butanol, and tetraethoxysilane (TEOS).

All chemicals were purchased from Sigma Aldrich. P123 (36 g) was mixed in 1302 mL of MilliQ water and 60 mL hydrochloric acid (HCl) under stirring at 35 °C. When the solution became completely transparent, 44.4 mL of butanol was added. After 1 h, 77.4 g of TEOS was inserted, and the solution was first left under stirring for 24 h. Subsequently, the solution was transferred in a closed propylene bottle at 35 °C or 100 °C for 24 h under static conditions. The temperature of the hydrothermal step influenced the pores size distribution, with the lower temperature (35 °C) resulting in the smallest

pores. After filtration, the silica was calcined at 550 °C for 10 h to remove the organic residuals. The resulting materials were named as KIT-6-35 °C and KIT-6-100 °C, depending on the temperature of the hydrothermal step.

### 2.1.2. Mesoporous CMK-8 and N-CMK-8 Carbon Materials

Ordered mesoporous carbons were produced using KIT-6 silica as a template. Furfuryl alcohol and furfuryl amine were used as carbon precursors for the production of CMK-8 and N-CMK-8, respectively. KIT-6-100 °C was used as a template for CMK-8 and KIT-6-35 °C as a template for N-CMK-8. All chemicals were purchased from Sigma Aldrich. Silica (KIT-6, 2 g) was impregnated with 2 mL of precursor (the quantity depended on the pore volume determined with BJH (Barret–Joyner–Halenda) method). First, a polymerization process was carried out in an air atmosphere at 100 °C for 2 h. The resulting material was further impregnated with half the amount of precursor and polymerized under air at 160 °C for 2 h. Pyrolysis was performed under $N_2$ flow at 950 °C for 2 h. Finally, the silica template was removed by washing in 1:1 MilliQ water and ethanol solution containing 2 M solution of NaOH.

### 2.2. Structural Characterization

$N_2$ adsorption and desorption isotherms were recorded at −196 °C using a Micromeritics Tristar instrument. The pore volume and pore size distribution were determined from the adsorption isotherm and the BJH (Barret–Joyner–Halenda) method, and the specific surface area was calculated by the BET (Brunauer-Emmet-Teller) multipoint method. The SAXS measurements were performed on a Mat:Nordic instrument from SAXSLAB. The XPS measurements were performed on a PHI 5000 Versa Probe III Scanning XPS Microprobe from ULVAC-PHI, Inc. SEM images were captured with a Leo Gemini ULTRA 55, ZEISS microscope.

### 2.3. Electrode Preparation

Electrodes were prepared by mixing mesoporous CMK-8 or N-CMK-8 with carbon black, Super P™ (Alfa Aesar, Haverhill, MA, USA), and Kynar® PVDF binder (Arkema, Colombes, France) in NMP (N-methyl-2-pyrrolidone, anhydrous, 99.5%, Sigma Aldrich AB, Stockholm, Sweden). The mixture was transferred to a glass vial, and Super-P™ and Kynar® 5% (*w/w*) solution in NMP were added. The solution was stirred overnight in a capped vial, forming a slurry having a weight ratio of (CMK-8/N-CMK-8: Super-P: PVDF) of (8:1:1). The slurry was coated onto a copper foil with a TQC AB3400 motorized automatic film applicator. A stainless-steel doctor blade (Wellcos Co., Seoul, Korea) was used to obtain a 250 μm thick coating of the slurry. The electrode sheets were dried for 1 day at ambient conditions in a fume hood, followed by drying in a vacuum oven at 80 °C and 20 mbar for 2 h.

### 2.4. Electrochemical Analysis and Device Testing

The electrochemical response of the CMK-8 and N-CMK-8 materials was tested as an anode in a Li-ion half-cell. The electrodes (~10 mm diameter, 2.5 mg·cm$^{-2}$ active material) were transferred to a glove box ($H_2O$ and $O_2$ less than 1 ppm) and dried under vacuum at 80 °C in a Buchi oven. Coin cells were assembled in an argon-filled glovebox using CR2032 housings. Glass microfiber (Sigma Aldrich) was used as the separator, soaked with 30 μL of commercial electrolyte LP30 (Sigma Aldrich), while a lithium metal disc was used as an anode (diameter 11 mm, areal density 4 mg·cm$^{-2}$). Galvanostatic charge-discharge tests were performed between 0.01 V and 2.9 V using a Scribner 580 battery cycler. The full-cells were prepared using the N-CMK-8 anode with a commercial LFP (LiFePO$_4$) cathode material. The lithium-ion cell was assembled and properly balanced by a positive/negative capacity ratio of 1:1. The mass loading of the carbon electrodes was about 1.5 mg·cm$^{-2}$.

## 3. Results and Discussion

### 3.1. Structural Characterization: $N_2$ Sorption, XRD, SAXS, and XPS Analysis

Figure 1a,c show the nitrogen adsorption–desorption isotherms of CMK-8 and N-CMK-8, respectively. The isotherms confirm the synthesis of mesoporous materials, displaying type-IV isotherms with what looks like a combination of H2 and H4 type hysteresis following the IUPAC classification [32]. This means that the materials exhibit a combination of spheroidal- and slit-shaped pores. The differences between the isotherms of the two samples indicate that their pore shape and/or pore tortuosity differ somewhat, where the N-CMK-8 has a broader pore size distribution. Based on Brunauer–Emmett–Teller (BET) analysis, the specific surface areas were 1240 $m^2$/g and 970 $m^2$/g for CMK-8 and N-CMK-8, respectively. The corresponding pore size distributions in Figure 1b,d, calculated by the Barret–Joyner–Halenda (BJH) method, further confirm the porosity difference. The CMK-8 shows a narrow pore size distribution peak at 4 nm, while the nitrogen functionalized N-CMK-8 was characterized by a trimodal porosity with three distribution peaks, the first at 4.2 nm, the second at 6.2 nm, and the third at 9.2 nm. The two different silicas, KIT-6-100 °C for CMK-8 and KIT-6-35 °C for N-CMK-8, were used to produce carbons with similar BET and pores size distribution. The lower specific surface area and the different pore size distribution of the N-CMK-8 were influenced by the different polymerization reactions and pyrolysis of the furfuryl amine precursor during synthesis [33,34]. Further explanation will be reported in the SAXS section.

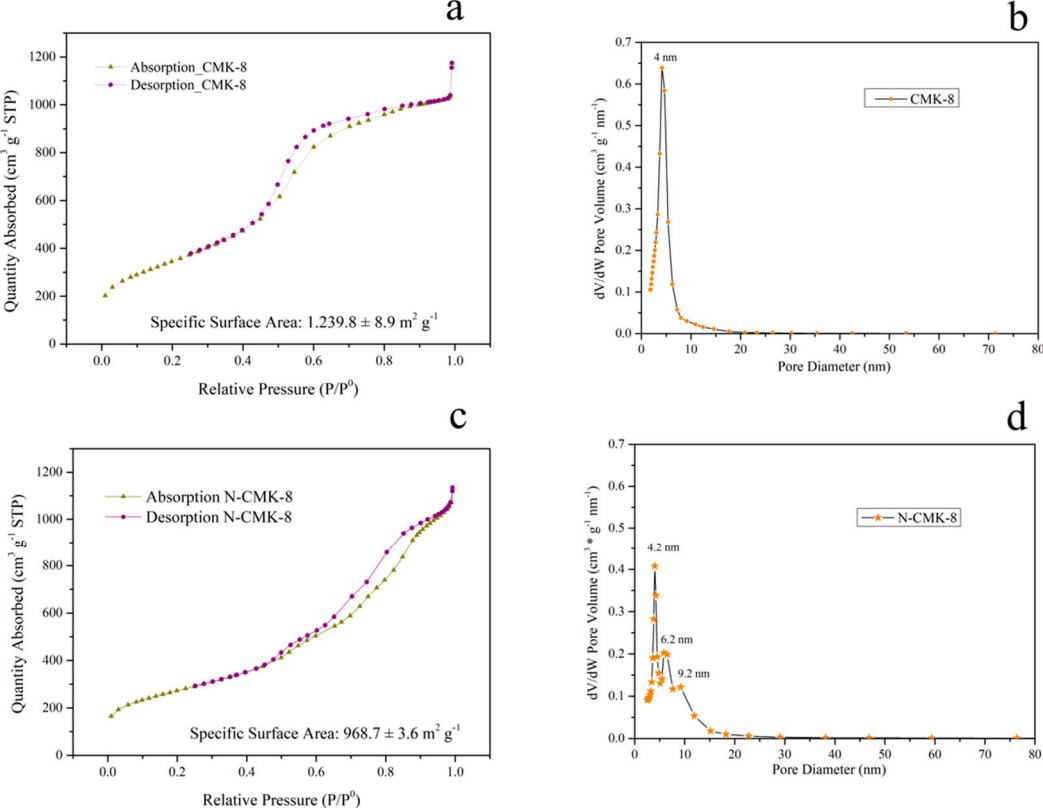

**Figure 1.** Barret–Joyner–Halenda (BJH) sorption isotherms (with $P^0$ = vapor pressure of the adsorbate) and pore size distributions of CMK-8 and N-CMK-8. (**a**) Absorption and Desorption Curves of CMK-8, (**b**) Pore size distribution of CMK-8. The distribution peak is centered at 4 nm, (**c**) Absorption and Desorption Curves of CMK-8, (**d**) Pore size distribution of N-CMK-8. The distribution peaks are centered at 4.2, 6.2 and 9.2 nm.

In the wide X-ray diffraction patterns of the CMK-8 and N-CMK-8 powders, reported in Figure A1, the materials showed two broad peaks ~25 and ~43. Those peaks were related to (002) and (100) diffractions, respectively. However, the wide-angle XRD was not resolutive for ordered, structured carbons as CMK8. Therefore, the materials were characterized by small angle X-ray scattering (SAXS).

Figure 2 presents the indexed SAXS patterns for the two carbons. The CMK-8 carbon shows peaks indexed as reflections from (211), (220), (321), (400), (420), and (332) planes, consistent with the cubic space group Ia-3d [35]. The pattern confirms the formation of highly ordered mesoporous carbon as a negative replica of cubic Ia-3d mesoporous silica (KIT-6) [36]. The N-CMK-8 carbon shows a somewhat different pattern, with peaks indexed as reflections from (110), (211), (220), (222), (321), and (400) planes. The appearance of the (110) and the small (222) peaks suggest the existence of a large fraction of displaced or uncoupled gyroidal sub-frameworks, and a lower level of symmetry, consistent with the tetragonal space group I41/a. This mesophase transition is associated with a process of strain relaxation upon silica removal [37–40]. The absence of a significant (110) peak in the CMK-8 pattern shows that the material is dominated by the undisplaced double gyroidal frameworks with Ia-3d symmetry, but the minute (110) peak visible indicates the presence of a small fraction of a lower symmetry mesophase. The CMK-8 keeps the Ia-3d symmetry thanks to the presence of complementary pores in the KIT-6 silica produced at 373 K, which connect the enantiomeric pair of 3D mesoporous channel systems. Upon pyrolysis of the carbon precursor, rigidly interconnecting carbon bridges avoid the structure displacement during silica removal. For N-CMK-8, the starting KIT-6 silica was synthesized at 308 K, where no complementary pores are typically formed. For this reason, the two carbon sub-frameworks produced during pyrolysis with this silica are displaced upon silica removal, and the structure transforms from the body-centered cubic Ia-3d to the body-centered tetragonal I41/a. The trimodal porosity shown by the N-CMK-8 seems to be associated with the symmetry modification. The first peak (4.2 nm) is related to the undisplaced Ia-3d domain. The structural displacements upon silica removal likely induce a distortion of the pores and the appearance of the second and third peak (6.2 nm and 9.2 nm).

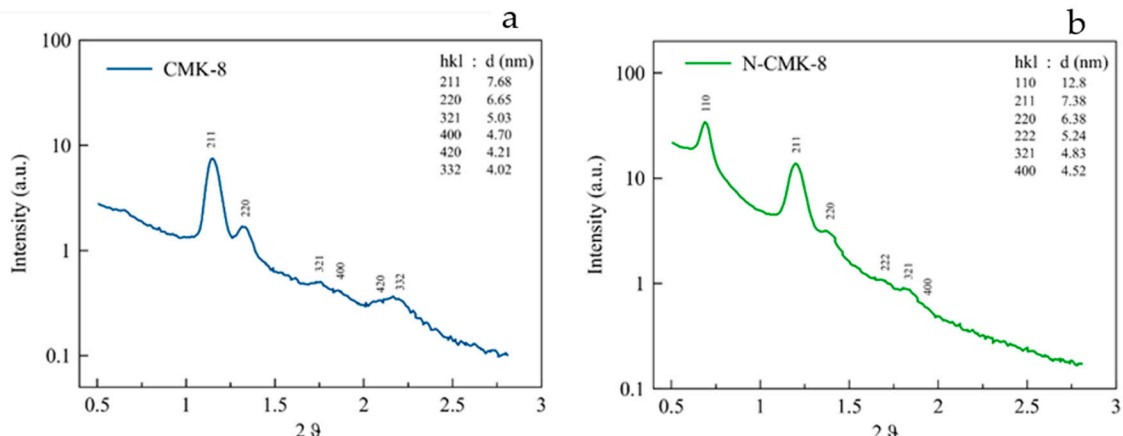

**Figure 2.** Small angle X-ray scattering (SAXS) patterns for CMK-8 and N-CMK-8. (**a**) SAXS pattern of CMK-8, (**b**) SAXS pattern of N-CMK-8. In both a and b panels, the peaks are indexed as reflections from crystallographic planes and reported in the table on the top right of the figures.

XPS was used to analyze the binding character of the N heteroatoms in the N-CMK-8 material, and the XPS spectra for the N-CMK-8 are reported in Figure 3. The three main peaks of the survey spectrum correspond to C 1s, O 1s, and N 1s, with a relative surface elemental composition of 81.2%, 12.2%, and 6.4%, respectively. The C 1s peak can be deconvoluted in three different components. The main peak at 284.4 eV is associated with $sp^2$ hybridized carbon or graphitic carbon. The peak at 285.7 eV is related to the bonding between carbon and nitrogen, indicating an N-$sp^2$ C bond. The third broad peak centered at 286.8 eV is not as identifiable, as it can have contributions from both N-$sp^3$ C

bonds and C-O bonds. Three sub peaks for C-O can be suggested, one at 286.3 eV for C-OH, one at 287.5 eV for C=O, and the last at 288.5 eV associated with carboxylic groups. Therefore, the carbon atoms constitute a honeycomb structure, which is modified by the nitrogen and oxygen incorporation. The high-resolution N 1s peak allows a better understanding of how the nitrogen is incorporated into the structure. The peak can be deconvoluted into three main sub-peaks at 398.4 eV, 399.4 eV, and 400.8 eV [41]. The first peak corresponds to pyridinic nitrogen, which substitutes a carbon atom on a C6 ring at the edge of the layer, and it is bound to two $sp^2$ carbons. The second peak is associated with pyrrolic nitrogen, which substitutes a carbon atom on the C5 ring with more $sp^3$ character. These two types of nitrogen contribute with electrons to the graphitic carbon layer, inducing donor states near the Fermi level [42]. The last peak at 400.8 eV corresponds to graphitic nitrogen, substituting a graphitic carbon and forming bonds with three $sp^2$ carbons. The high-resolution O 1s peak is more complicated to describe, considering all the different contributions. We suggest a deconvolution with four sub peaks: The first at 530.5 eV is associated to C≡O in quinone; the second at 531.8 eV to oxygen in hydroxyls, ethers, C=O in esters, amides and anhydrides; the third at 533.4 eV to C-O in esters and anhydrides and contributions from carboxylic groups; and the fourth at 535.6 eV is attributed to adsorbed water [43].

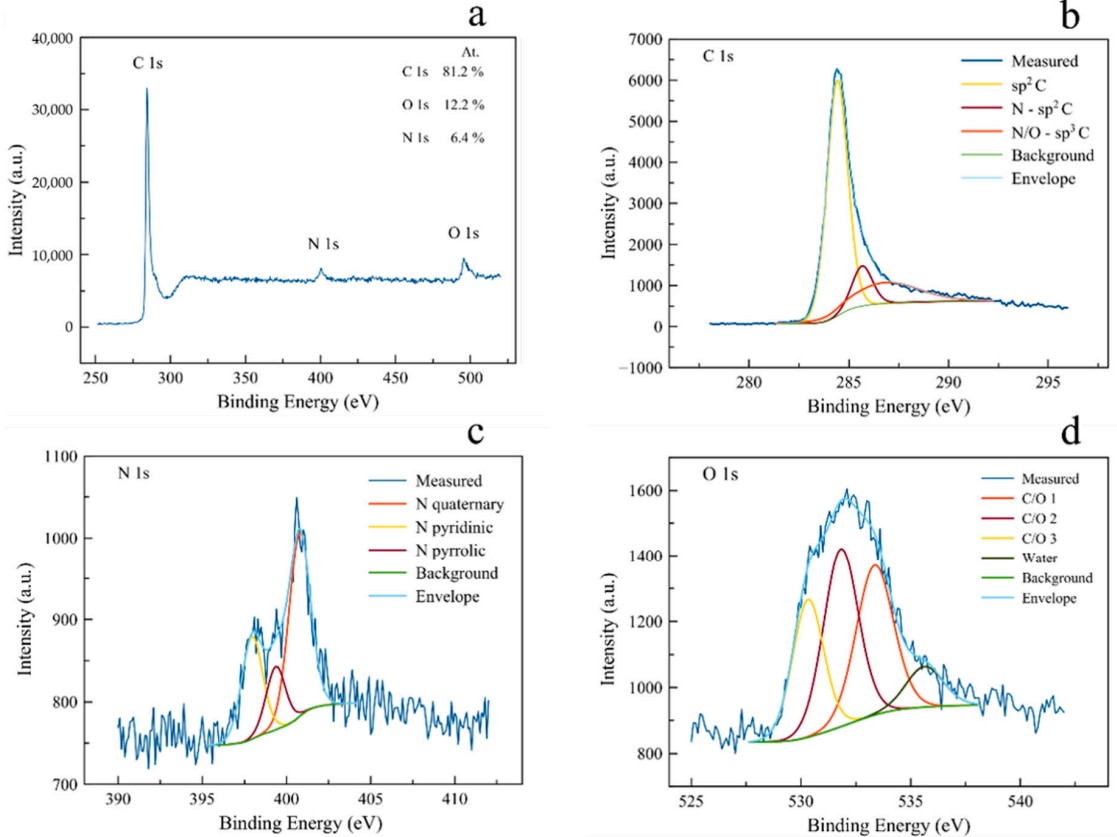

**Figure 3.** XPS spectrum of N-CMK-8 with relative surface elemental composition. (**a**) Full spectrum; (**b**) high-resolution of C 1s, (**c**) high-resolution of N 1s, and (**d**) high-resolution of O 1s.

### 3.2. Electrochemical Evaluation

The ordered mesoporous carbons were studied in a half-cell configuration vs. Li metal. SEM images of the composite electrode (CMK-8: Super P: Kynar®) are presented in Figure A2. Figure 4a,b report the voltage profiles vs. the specific capacity of the N-CMK-8 and CMK-8, respectively. Both materials show a very high irreversible charging capacity between the first and second cycles. The irreversibility is connected to the extended plateau starting at around 1 V in the first cycle, which is associated with electrolyte decomposition at the surface of the active material and formation of a solid electrolyte

interface (SEI) [44]. The presence of edges and defects in high surface area materials provides many sites for the electrolyte decomposition, which explains the large irreversible capacity. The introduction of nitrogen and formation of the pyridinic and pyrrolic sites further increases the side reactions and Li-ion trapping, resulting in an even higher irreversible capacity compared to the CMK-8 without nitrogen. However, the presence of nitrogen has also a very beneficial effect in terms of electrochemical performance. As seen in Figure 4a, the N-CMK-8 can reversibly store ~1000 mAh·g$^{-1}$, which is more than double than the available reversible capacity of the CMK-8. The numerous nitrogen defects at the high surface area act as Li insertion sites, enhancing the storage capability. Moreover, the low energy barrier for Li insertion and adsorption to the pyridinic and pyrrolic defects provides a higher specific capacity [45].

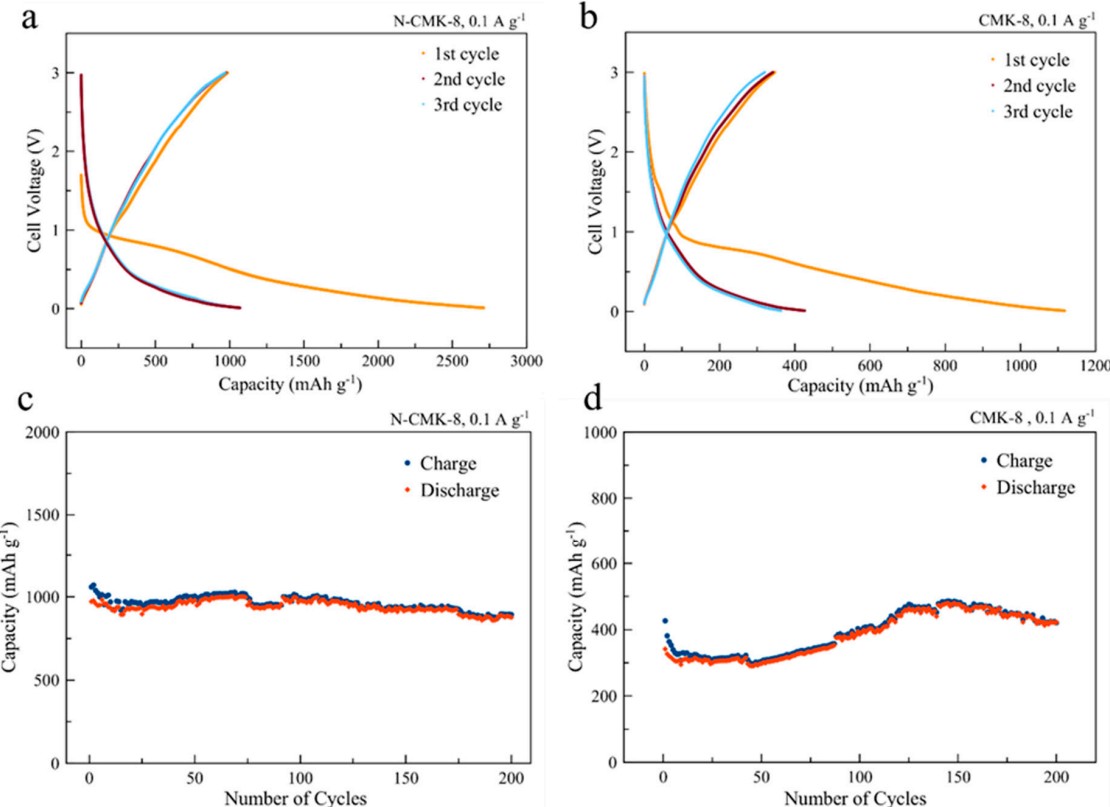

**Figure 4.** Voltage profiles for the first 3 cycles of (**a**) N-CMK-8 and (**b**) CMK-8. Long-term specific capacities of (**c**) N-CMK-8 and (**d**) CMK-8.

Figure 4c,d report the charge and discharge capacities vs. cycles number at 0.1 A·g$^{-1}$ for the two carbons. The increase in capacity of the CMK-8, after the initial 10 cycles of irreversible formation of the SEI, is likely due to a better infiltration of the electrolyte inside the mesopores of the material as cycling proceeds. For the N-CMK-8, the SEI formation is reduced, probably due to a combination of the broader porosity and the increased polarity of the nitrogen-functionalized surface. The N-CMK-8 shows very good reversible performance and stability for 200 cycles, being competitive compared to different N-doped carbon materials reported in the literature (see Table 1).

**Table 1.** CMK-3 and N-doped carbon materials used as anodes in Li-ion batteries are compared with this work (N-CMK-8). Potential windows, specific capacity, capacity retention, and relative reference is listed for each material in the table.

| Material | Potential Window (V) | Specific Capacity at 0.1 A·g$^{-1}$ (mAh·g$^{-1}$) | Capacity Retention (Number of Cycles) | Ref. |
|---|---|---|---|---|
| This work | 0.01–3 | ~950 | ~95% (200) | |
| CMK3 | 0.01–3 | ~400 | ~88% (100) | [21] |
| N doped carbon nanotubes | 0.01–3 | ~600 | ~93% (10) | [16] |
| N doped carbon nanofibers | 0.01–3 | ~850 | >95% (150) | [46] |
| N doped graphene nanosheets | 0.01–3 | ~720 | ~100% (10) | [45] |
| N doped porous carbon nanofiber webs | 0.01–3 | ~1280 | >100% (600) (2 A·g$^{-1}$) | [47] |
| N doped double shelled hollow nanospheres | 0.01–3 | ~850 | ~100% (100) | [48] |

The charge-discharge rate performance of the N-CMK-8 is reported in Figure 5a. The electrode shows good behavior at high-rate, delivering ~200 mAh·g$^{-1}$ at 5 A·g$^{-1}$ and showing a nearly unaltered performance when cycled again at 0.1 A·g$^{-1}$. The mesoporous structure provides short diffusion paths for the Li ions penetration and a desirable high surface to volume ratio. Besides, the low adsorption energies for Li of the nitrogen functionalities presumably enhance the charge transfer reactions.

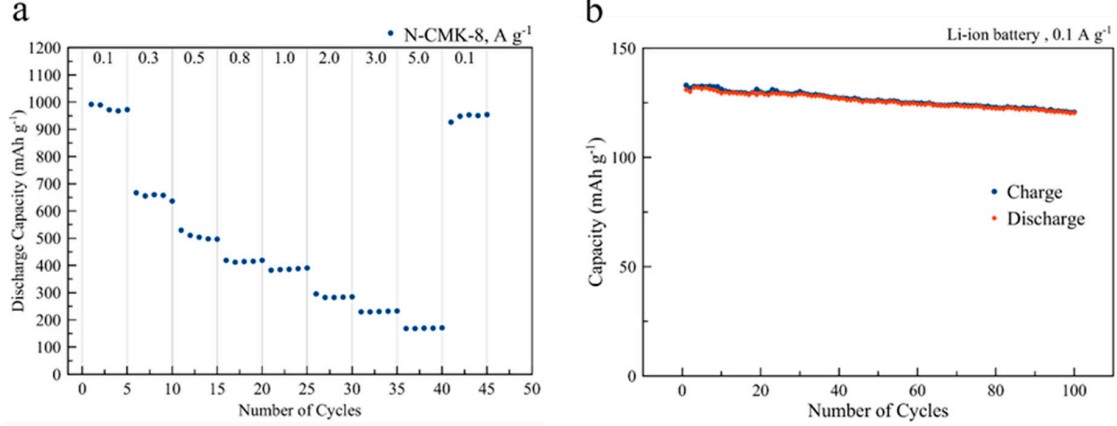

**Figure 5.** (**a**) Rate capability of N-CMK-8 and (**b**) long term capacity of the Li-ion cell.

Finally, the N-CMK-8 anode was tested in combination with a LiFePO$_4$ cathode in a full cell. LiFePO$_4$ is typically used in laboratory studies as a cathode because of its environmental friendliness, but it has a lower specific capacity compared to NMC (Lithium Nickel Manganese Cobalt Oxide) cathodes used nowadays in commercial applications [49,50]. Figure 5b shows that the Li-ion cell has stable performance over 100 cycles (>90% capacity retention [51]) with high coulombic efficiency, demonstrating the feasibility of using nitrogen functionalized ordered mesoporous carbons as anodes for practical applications in LIBs.

## 4. Conclusions

Ordered mesoporous carbon of type CMK-8 and nitrogen-functionalized CMK-8 have been synthesized and structurally characterized. The N-CMK-8 exhibits a broader pore size distribution with three distinct pore size distribution peaks. High-resolution XPS of the N 1s spectrum showed that differently bound nitrogen atoms (pyridinic, pyrrolic, and graphitic) are present in the N-CMK-8 structure. Galvanostatic charge and discharge (GC/D) curves show the beneficial effect of the nitrogen

on the storage capability of an N-CMK-8 anode. The nitrogen defects appear to provide a lower energy barrier for lithium insertion and adsorption.

In half-cell configuration, the N-CMK-8 can reversibly store ~1000 mAh·g$^{-1}$ at 0.1 A·g$^{-1}$ and retain ~200 mAh·g$^{-1}$ at the fast rate of 5 A·g$^{-1}$. A full cell with LiFePO$_4$ as cathode material and LP30 as electrolyte shows stable performance over 100 cycles with very high coulombic efficiency, comparable to similar commercial cells. Our results demonstrate that controlling the porosity and functionalizing mesoporous carbons such as CMK-8 is a highly promising strategy to improve anode performance for Li-ion batteries of the future.

**Author Contributions:** G.C. and C.C. have carried out the synthesis and the characterizations of the CMK-8 and N-CMK-8. G.C. and M.A. have built the cells and performed all the electrochemical characterization. S.X. carried out the XPS measurements and analyzed the data. G.C. and C.C. wrote the paper. A.M. and A.E.C.P. revised the paper and discussed the results. All the authors contributed to the discussion and editing of the manuscript draft. All authors have read and agreed to the published version of the manuscript.

**Funding:** This work has been carried out with support from the Swedish Energy Agency project number 39045-1.

**Acknowledgments:** The authors want to acknowledge Yu Cao, Associate Professor @ Chalmers University of Technology, for the XPS measurement. The authors acknowledge the support from the Chalmers Areas of Advance Materials Science and Energy, the Swedish Energy Agency, and the access to microscopy facilities of Chalmers Materials Analysis Laboratory.

**Conflicts of Interest:** The authors declare no competing interests.

## Appendix A

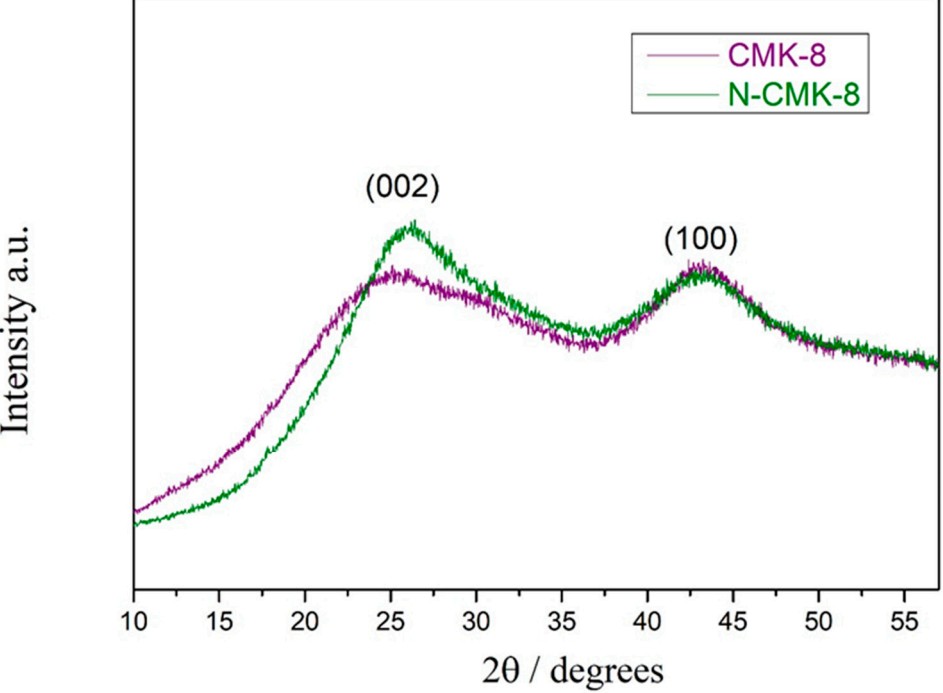

**Figure A1.** X-ray diffraction patterns of CMK-8 (purple line) and N-CMK-8 (green line). The peaks are indexed with the respective Miller index.

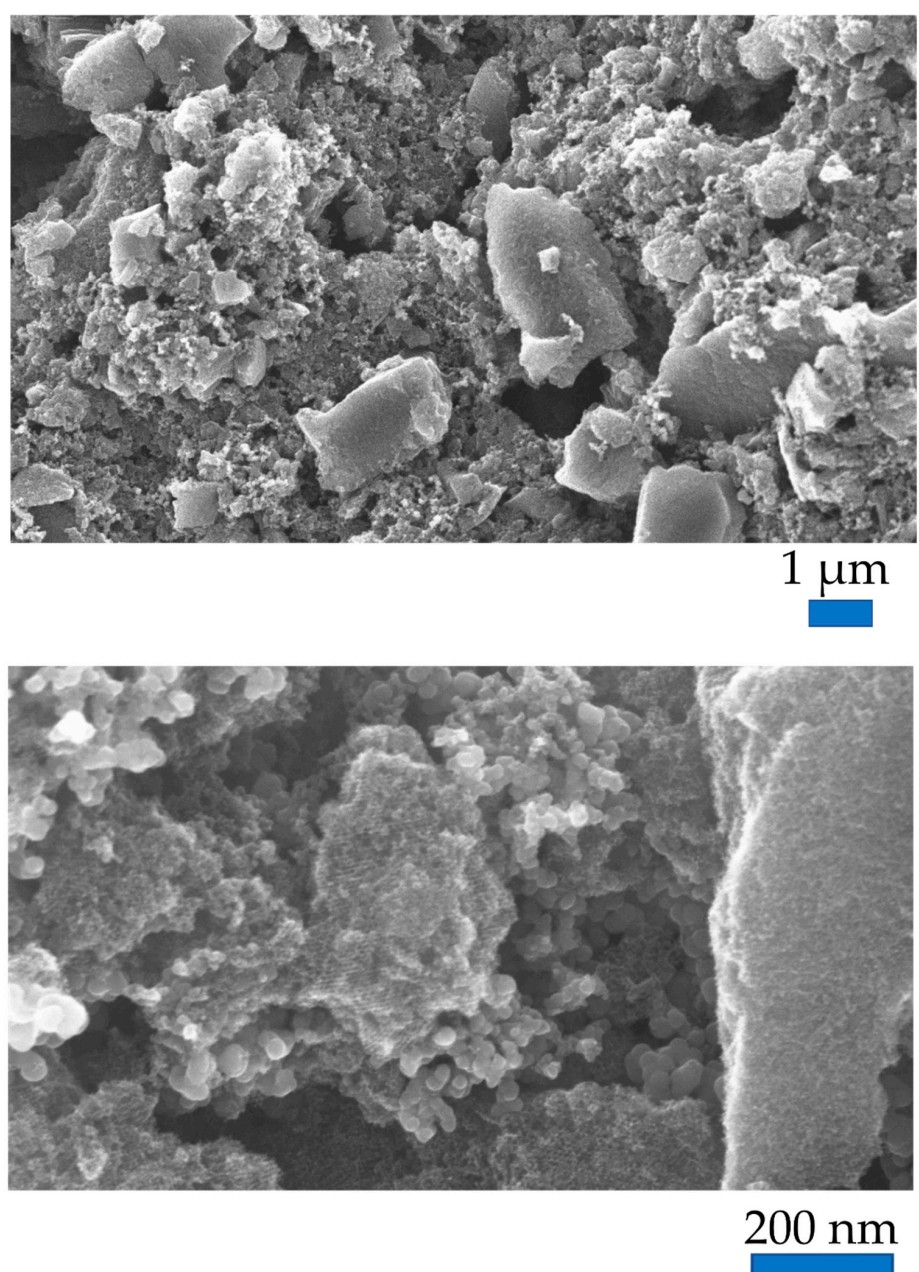

**Figure A2.** SEM micrographs of the N-CMK-8 composite electrodes.

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
