# Peer review of "Effect of Nitrogen Doping on the Performance of Mesoporous CMK-8 Carbon Anodes for Li-Ion Batteries"

_energies, doi:10.3390/en13194998_

Round 1

Reviewer 1 Report

In their Manuscript, the Authors produced new N-doped mesoporous carbon anode material. Moreover, they prepared and tested a full Li-ion cell based on the new material. The conducted study are urgent, well-designed and potentially high-impact. I recommend the paper for publication, but the following minor comments should be accounted:

1) According to Fig. 1d, N-doped material possesses three peaks in pores size distribution. Some explanations why nitrogen doping results in these peaks appearing should be added. Are their positions depend on temperature during the material fabrication?

2) Caption to Fig.1 should be extended (what gas was adsorbed (N2?), what is p0 (atmosphere pressure?)).

3) Figures are very unclear and have low resolution. Technical improving of their quality is needed.

Author Response

Dear Reviewer,

please, find the file attached as response to your comments.

Best regards,

Carmen Cavallo

Reviewer 2 Report

The current manuscript demonstrates the effect of Nitrogen doping Mesoporous CMK-8 Carbon Anodes for the Li ion battery performance. The key finding of the article is suitable for this journal as well as it would be beneficial for the readers working in the field of Li ion batteries. However, few minor drawbacks are there in this article. Therefore, I recommend minor revision of this manuscript. My critical comments are given below –

  1. A comparison table of the N-doped anode materials with other N-doped carbon materials should be included in the manuscript.
  2. The figures are not clearly visible throughout the manuscript. The authors should provide figures with clear resolution.
  3. The XRD analysis of the anode material should be done.
  4. What is the doping level of the doped anode material? The doping level should be included in the manuscript.
  5. In the introduction part, the authors should cite few recent articles related to the carbon-based electrodes for Li-ion battery as well as other energy storage application such as 1. Progress in Energy and Combustion Science 75, 100786, 2019, 2. Carbon, 94, 455-463, 2015, 3. Nano-Micro Letters, 10, 21, 2018, 4. Chemical Engineering Journal, 381, 122588, 2020, 5. Electrochimica Acta, 334, 135619, 2020, 6. Nano Research 12 (11), 2655-2694, 2019, 7. Materials Today, DOI: 10.1016/j.mattod.2020.04.010, 2020, 8. Electrochimica Acta, 342, 136115, 2020, 9. Journal of Alloys and Compounds, 831, 154607, 831, 10. Electrochimica Acta, 359, 136924.

Author Response

Dear Reviewer,

please, find the attached file as response to your comments.

With my best regards,

Carmen Cavallo

Reviewer 3 Report

This paper studied effects of nitrogen doping of ordered porous carbon as anodes for Li-ion batteries. As claimed by the authors, present data support facts that nitrogen doping contributed to enhancement of cyclability and capacities. Full cell batteries test could be a good reference for this filed. However, this type of research using porous carbons has been performed by many groups, so it is difficult to find any novelty. Below is minor comments on this paper

  1. Pp 6, line 218, there is a typo (Figure 4 b and 4c) -> (Figure 4c and 4d)
  2. At the same line, why CMK-8 (Figure 4d) showed slight increase in capacity after 100 cycles?
  3. Can you provide any comparison with other reported data? Making a table would be good.
  4. In the full cell test, can you say that 100 cycles stability is sufficient for commercialization or having potential for field application?
  5. Please describe future direction to enhance stability using N-doped ordered porous carbon. 

Author Response

(The authors gave the same response as above.)
